# Daily Physical Activity and Sleep Measured by Wearable Activity Trackers during the Coronavirus Disease 2019 Pandemic: A Lesson for Preventing Physical Inactivity during Future Pandemics

**Hidetaka Hamasaki** 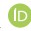

Hamasaki Clinic, 2-21-4 Nishida, Kagoshima 890-0046, Japan; h-hamasaki@umin.ac.jp; Tel.: +81-099-250-3535

**Abstract:** Wearable activity trackers are devices that are comfortably worn on the body and are designed to be effective in monitoring daily physical activity and improving physical fitness of the wearer. This review aimed to investigate the impact of the coronavirus disease 2019 (COVID-19) pandemic on physical activity measured using wearable activity trackers and discuss future perspectives on wearable activity trackers during pandemics. Daily physical activity was significantly decreased during the COVID-19 pandemic. The implementation of strict public health measures, such as total lockdown, can decrease people's physical activity by 50% or more of that prior to the lockdown. Physical inactivity is significantly associated with several health problems, including obesity, diabetes, cardiovascular disease, and cancers; therefore, an effective healthcare system to prevent physical inactivity during pandemics should be established. It is essential to create a network between healthcare organizations and wearable activity tracker users to monitor real-time health status and prepare for the future pandemic.

**Keywords:** wearable device; smartphone application; telemedicine; physical activity; COVID-19

## 1. Introduction

The coronavirus disease 2019 (COVID-19) pandemic has emerged and become prolonged, significantly affecting the lives of people worldwide. To date (September 2021), over 220,000,000 coronavirus cases and 4,500,000 deaths due to COVID-19 have been confirmed across the globe [1]. New coronavirus variants have been appearing one after another, and some mutated strains, such as the Delta variant, may be more contagious, virulent, and vaccine-resistant, which will make it difficult for humans to manage the pandemic effectively [2]. Several countries took non-pharmaceutical public health measures such as lockdown [3] and social distancing [4] in addition to conventional infection control measures (e.g., handwashing and wearing of face masks); however, the spread of COVID-19 is not currently controlled. People are required to make behavioral and lifestyle changes in this pandemic era. The development of technologies and systems that enable disease prevention, effective healthcare without a risk of infection, and protection of human life is critical [5].

Wearable devices that can comfortably be worn on the body and perform several tasks in conjunction with handheld devices, such as smartphones, play a pivotal role in healthcare [6]. Recent systematic reviews have shown that wearable devices are useful for monitoring heart rate and sleep in hospital settings [7], increasing physical activity in children and adolescents [8], improving health-related outcomes in patients with cancer [9], and reducing body weight in individuals with obesity [10]. In addition, during the COVID-19 pandemic, sleep pattern and duration were accurately and effectively monitored by wearable devices [11–15]. On the other hand, social distancing or self-isolation can decrease daily physical activity [16], and such physical inactivity may increase cardiovascular risks [17]; further, physical inactivity itself was associated with an increased risk

of hospitalization due to COVID-19 (relative risk = 1.32; 95% confidence interval [CI], 1.10–1.58) [18]. Physical inactivity is a global health problem responsible for the increasing risk of noncommunicable diseases, such as diabetes, coronary heart disease, and breast and colon cancers, and Lee et al. [19] reported that the life expectancy of humans would increase by 0.68 years by increasing physical activity. Promoting and increasing physical activity are the cornerstone of the management of diabetes and obesity, which are now the risks of severe COVID-19 as well as public health epidemics. The aim of this review is to summarize the current evidence regarding the impact of the COVID-19 pandemic on physical activity and sleep measured by using wearable activity trackers and discuss future perspectives of wearable activity trackers during pandemics.

## 2. Physical Activity during the COVID-19 Pandemic

In this narrative review, the author focuses on the impact of the COVID-19 pandemic on physical activity measured using wearable activity trackers. The author performed a literature search on COVID-19 and wearable devices in PubMed. The search terms were "COVID-19" and "wearable". The titles and abstracts of the identified articles were reviewed to determine their relevance. The author excluded editorials, commentaries, and letters. The search of PubMed from its inception to August 2021 yielded 253 articles. Of these, nine studies were included in this review (Figure 1).

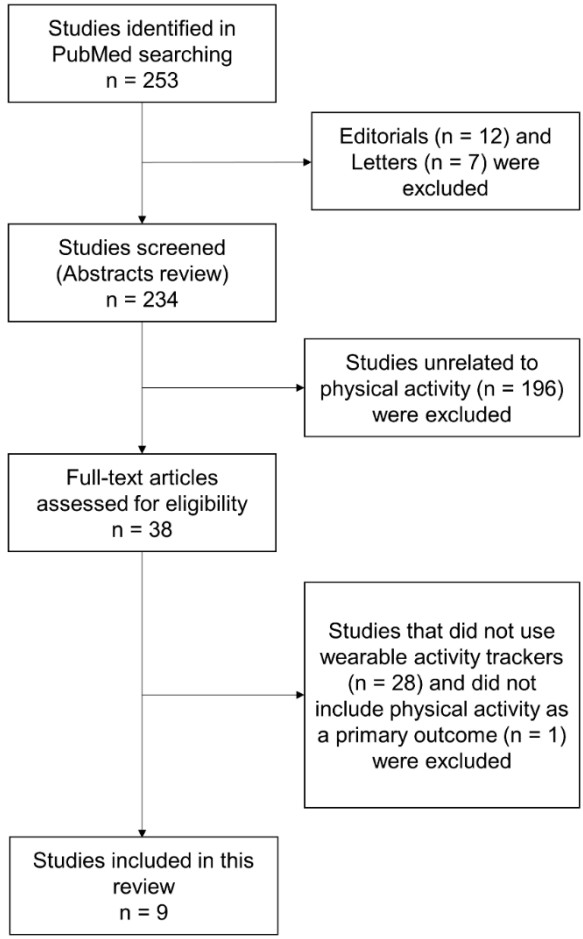

**Figure 1.** Review flow chart.

The Remote Assessment of Disease and Relapse–Central Nervous System (RADAR-CNS) Consortium reported that people in Italy, Spain, Denmark, the United Kingdom (UK), and the Netherlands significantly changed their behavior during the COVID-19 pandemic based on the data from wearables and mobile technologies [20]. A total of 1062 subjects were

recruited in the RADAR-CNS study, and the following data were collected through a smartphone or a Fitbit device on a 24/7 basis: smartphone location (maximum traveled distance from home and homestay), smartphone Bluetooth (maximum number of nearby devices), Fitbit step count, Fitbit sleep (bedtime and sleep duration), Fitbit heart rate (average heart rate), smartphone user interaction (unlock duration), and smartphone use event (social application use duration). Subjects spent more time on social media (Italy, Spain, the UK, and The Netherlands), had a lower heart rate (Italy, Spain, and Denmark), slept later (Italy, Spain, the UK, and The Netherlands), and slept longer (Italy, Spain, and the UK) during the lockdown than before the lockdown. Young subjects (<45 years) also had fewer daily steps during the lockdown than before the lockdown (Italy, the UK, and The Netherlands). Subjects whose body mass index (BMI) was less than 25 kg/m$^2$ walked more than subjects with a high BMI ($\geq$25 kg/m$^2$). Non-pharmaceutical measures such as lockdown made people inactive, and the health of people with overweight/obesity could be damaged more by lockdown than people with normal weight. Interestingly, there were different results across countries; for example, Denmark showed small behavioral changes in response to the lockdown. Apart from the effectiveness of lockdown for infection control during the pandemic, young people with obesity should be monitored carefully to prevent health problems due to physical inactivity when implementing lockdown and social distancing. Importantly, obesity is significantly associated with the increased risk of susceptibility to COVID-19 (odds ratio (OR), 2.42; 95% CI, 1.58–3.70), severity of COVID-19 (OR, 1.62; 95% CI, 1.48–1.76), hospitalization (OR, 1.75; 95% CI, 1.47–2.09), and death (OR, 1.23; 95% CI, 1.06–1.41) [21].

A small retrospective observational study in Poland showed similar results [22]. Five subjects who self-isolated at home during the COVID-19 pandemic were enrolled in this study, and their daily steps, resting heart rate, and sleep duration were recorded for 464 days. The data were collected through Fitbit Versa smartwatches. The self-isolation period was 50 days. The daily step count (from 7550 $\pm$ 4430 to 3230 $\pm$ 1910 steps) and resting heart rate (from 63.28 $\pm$ 4.36 to 60.23 $\pm$ 3.69 beats per minute (bpm)) were significantly decreased during the self-isolation period compared with those before the lockdown, but no significant change was noted in sleep duration. However, sleep duration (from 429.64 $\pm$ 66.80 to 371.73 $\pm$ 117.19 min) was significantly decreased in subjects aged $\geq$80 years during the self-isolation period. The cardiovascular and pre-frailty risk assessment score using age, daily steps, resting heart rate, and sleep duration was developed in this study; four out of five subjects increased the score during the self-isolation period, and two out of five subjects were considered to have an increased cardiovascular and pre-frailty risk due to self-isolation. This study suggests that self-isolation is harmful for physical health and increases the risk of cardiovascular disease and frailty. The authors undoubtedly expect that smartwatches or other medical wearables will play an essential role in telemedicine when a public health emergency, such as a pandemic, occurs.

Mishra et al. [23] investigated the impact of social isolation during the COVID-19 pandemic on wellbeing in older adults. There were a total of 10 adults who were aged $\geq$75 years or $\geq$65 years with a risk of falling. Physical activity and sleep were monitored using a wearable pendant device (ActivePERS/PAMSys). Study subjects wore the wearable pendant device for two consecutive days, and posture (sitting%, standing%, lying%, and walking%), walking characteristics (daily step count, number of walking bouts, and cadence), postural transition, activity behavior (prolonged sitting and light- and moderate- to vigorous-intensity activities), and sleep quality (time in bed) were measured. The daily step count (from 5911 $\pm$ 1193 to 2655 $\pm$ 419 steps), number of walking bouts (from 241.3 $\pm$ 56.2 to 148.6 $\pm$ 27.1), number of postural transitions (from 720.7 $\pm$ 162.2 to 399.4 $\pm$ 68.5), standing% (from 16.5 $\pm$ 2.3% to 11.1 $\pm$ 1.8%), and walking% (from 6.7 $\pm$ 1.3 to 3.2 $\pm$ 0.5%) were significantly decreased, and sitting% (from 37.5 $\pm$ 4.5% to 45.2 $\pm$ 5.1%) was significantly increased during the COVID-19 pandemic compared with those before the pandemic. Furthermore, depression symptoms evaluated by the Center for Epidemiologic Studies Depression scale significantly increased from 3.0 $\pm$ 0.7 to 7.5 $\pm$ 2.4 during

the pandemic. However, sleep quality did not change. Depression symptoms were also correlated with deterioration in daily physical activity such as a decline in cadence and increase in time in bed and prolonged sitting time. Social isolation during the pandemic has a negative impact on daily physical activity and mental health in older adults.

Woodruff et al. [24] investigated how physical activity, sedentary behavior, and stress changed during the first month of the COVID-19 pandemic. Subjects who regularly used wearable activity trackers were recruited via social media, and 121 participants completed the study. Physical activity was objectively measured by wearable activity trackers manufactured by Apple, Fitbit, Samsung, and Garmin. The daily step count was decreased from $9509 \pm 3390$ to $8497 \pm 3620$ steps, whereas the sedentary time was increased from $181.79 \pm 117.9$ to $299.6 \pm 136.6$ min per day from pre- to post-COVID-19. However, 55 participants reported that their physical activity was increased; on the other hand, an equal number of participants reported that their physical activity was decreased, and 11 participants reported that their physical activity did not change. There seems to be a discrepancy between objectively measured physical activity and self-reported physical activity during the pandemic. Major reasons/barriers to changes in physical activity were access/equipment and motivation in participants whose self-reported physical activity was decreased and time in participants whose self-reported physical activity was increased during the pandemic. Daily and work stresses were significantly increased due to COVID-19, and a decrease in physical activity was associated with a larger work stress. These two studies assessed the influence of the pandemic on mental health of humans as well as physical activity, both of which indicate that the pandemic itself and/or physical distance measures, such as lockdown and self-isolation, negatively affect people's mental state. The COVID-19 pandemic is associated with anxiety, depression, and insomnia in the general population and healthcare workers, and this stress-related mental state is associated with suicidal behavior [25]. Wearable activity trackers may also play a useful role in monitoring daily physical activity in individuals who are at high risk of such mental illness.

Ong et al. [26] examined the impact of mobility restrictions during the COVID-19 pandemic on physical activity and sleep pattern using data from the Health Insights Singapore study. Physical activity and sleep duration were measured using a Fitbit. The daily step count on weekdays/weekends (from $9344 \pm 3634/8992 \pm 4437$ to $5284 \pm 4283/5432 \pm 4692$ steps, respectively), moderate- to vigorous-intensity activity minutes on weekdays/weekends ($37.6 \pm 27.9/40.9 \pm 37.5$ to $24.4 \pm 28.2/27.2 \pm 32.9$ min, respectively), and resting heart rate on weekdays/weekends ($65.7 \pm 7.3/65.8 \pm 7.3$ to $64.2 \pm 7.3/64.1 \pm 7.4$ bpm, respectively) were significantly decreased during the lockdown. Sleep duration on weekdays/weekends was increased from $6.92 \pm 0.95/7.49 \pm 1.18$ to $7.28 \pm 0.96/7.73 \pm 1.18$, respectively; however, sleep efficiency was not changed during the lockdown. These changes were proportional to the level of mobility restrictions.

French researchers examined whether wearable activity trackers could detect adherence to home confinement policies during the COVID-19 pandemic [27]. The daily step count was monitored using a wristwatch with an embedded accelerometer (Withings). Data from approximately 742,000 subjects were analyzed. Overall, the number of steps significantly decreased (7.8–55.6% of baseline steps per day) in countries with a total lockdown. However, a decrease in the number of steps in countries with a partial lockdown was 1.5–22.1% of baseline steps per day, and an increase (1.3% of baseline steps per day) in the number of steps was observed in Germany. In countries without lockdown, the number of steps also seemed to be decreased; however, it was significantly increased in Sweden (5.7% of baseline steps per day). These findings suggest that people adhered to the home confinement policies during the pandemic, but these raised an issue that such strong policies could harm cardiometabolic health. Although this study showed that wearable activity trackers were useful to detect adherence to the pandemic policies, the strategy to make use of data from wearable activity trackers for health management and prevent damage to cardiometabolic health is not exhibited.



Capodilupo and Miller [28] assessed changes in cardiovascular indicators of health as well as sleep and exercise behaviors due to physical distancing during the COVID-19 pandemic. A total of 5436 individuals who used a wearable device, the WHOOP strap, in the United States (US), were retrospectively analyzed. The WHOOP strap can automatically detect exercise, and the following variables were identified: sleep duration, sleep onset, sleep offset, social jet lag, exercise type, exercise frequency, exercise intensity, resting heart rate, and heart rate variability. Sleep duration was significantly increased from $7.65 \pm 0.20$ to $7.89 \pm 0.18$ h after the implementation of physical distancing. People have went to bed earlier and got up later. Exercise frequency was increased from $59.9 \pm 4.04\%$ to $61.91 \pm 4.36\%$ during physical distancing compared with baseline in 2020. However, exercise frequency was decreased from $65.62 \pm 10.44\%$ to $62.83 \pm 8.22\%$ in subjects aged 18–25 years. A large proportion of subjects were engaged in regular exercise and increased exercise intensity during the physical distancing period. The resting heart rate decreased from $55.09 \pm 0.81$ to $54.2 \pm 0.55$ bpm, and the heart rate variability increased from $65.85 \pm 2.28$ to $66.82 \pm 1.4$ ms during the physical distancing period in 2020; however, the increase in heart rate variability was also observed during the control period in 2019, and it was larger than that during the physical distancing period. The authors speculated that the increased exercise intensity during the physical distancing period may increase the effect of the sympathetic nervous function.

Zinner et al. [29] investigated the impact of social distancing and lockdown on daily physical activity, sleep, and training mode and intensity in highly trained athletes, kayakers, and canoeists in Germany. This retrospective observational study enrolled 14 young athletes, and their data concerning sleep duration, daily physical activity, and heart rate during each training session were collected using a multi-sensor smartwatch (Polar M430). Sleep duration was 30 (6.7%) min longer (from $451 \pm 22$ to $481 \pm 17$ min) during the lockdown. The time spent lying down was significantly increased (from $623 \pm 60$ to $729 \pm 21$ min; 17%), while the time spent on light-intensity physical activity (from $205 \pm 25$ to $190 \pm 5$ min; $-7.3\%$), time spent on moderate-intensity physical activity (from $59 \pm 15$ to $48 \pm 14$ min; $-18.6\%$), and sitting time (from $360 \pm 33$ to $326 \pm 21$ min; $-9.4\%$) were significantly decreased during the lockdown. The vigorous-intensity physical activity did not change between before and during the lockdown. Decreases in overall training time per week ($-27.6\%$) and average duration of each training session ($-15.4\%$) were also observed during the lockdown. The total training time spent at <60%, 82–88%, 89–93%, or 94–100% of the individual peak heart rate was 2.8–17.5% higher and that spent at 60–72% or 73–83% of the individual peak heart rate was 4.3–18.7% lower during the lockdown. Athletes were engaged in high-intensity training longer during the lockdown than before the lockdown; however, the training time and time spent on light- to moderate-intensity physical activity were decreased. Overall, athletes also became inactive during the lockdown.

Taylor et al. [30] quantified the effect of lockdown on physical activity in patients with heart failure. Physical activity data were collected from modern cardiac implantable electronic devices (CIEDs) in the Triage HF Plus Evaluation Study in the UK. CIEDs have multiple built-in sensors to monitor real-time physiological data including physical activity, thoracic impedance, heart rate, heart rhythm, and atrial arrhythmias. A total of 311 patients who completed the collection of 8-week activity data (pre-lockdown for 4 weeks and post-lockdown for 4 weeks) were enrolled in this study. Daily physical activity was measured by an accelerometer within the CIED. A total of 246 patients (79.1%) showed a reduction in physical activity post-lockdown. The median physical activity per day was significantly decreased from 134.7 min/day during pre-lockdown to 113.9 min/day during post-lockdown. Daily physical activity was immediately decreased within 2 weeks after the implementation of lockdown. There were no characteristics of patients such as age, sex, BMI, frailty, and severity of heart failure that were clearly associated with the reduction in physical activity due to the lockdown. Table 1 summarizes the studies investigating the change in daily physical activity during the COVID-19 pandemic.

**Table 1.** Changes in daily physical activity measured by wearable activity trackers during the COVID-19 pandemic.

| Authors, Year | Subjects Countries | Study Design Study Period | Wearable Activity Trackers | Results |
|---|---|---|---|---|
| Sun et al., 2020 [20] | 1062 patients with major depressive disorder or multiple sclerosis in Italy, Spain, Denmark, the United Kingdom, and Netherlands Age: No description BMI: No description | Prospective cohort study a part of the RADAR-CNS studies Between 1 February 2019 and 5 July 2020 | Smartphone Fitbit | Daily step count↓ in young subjects Heart rate↓ Time spent on social media↑ Sleep duration↑ |
| Kańtoch E and Kańtoch A, 2021 [22] | 5 adult volunteers (2 men and 3 women, 2 subjects with history of cardiovascular diseases) Poland Age: 57 ± 22.38 years BMI: 27.80 ± 2.95 kg/m$^2$ | Retrospective observational study Between 22 January 2019 and 30 April 2020 | Fitbit Versa smartwatch | Daily step count↓ Resting heart rate↓ Sleep duration→ |
| Mishra et al., 2021 [23] | 10 community-dwelling older adults (6 men and 4 women) United States Age: 77.3 ± 1.9 years BMI: 27.5 ± 1.6 kg/m$^2$ | Prospective observational study Between January-March 2020 and March–September 2020 | ActivePERS/PAMSys pendant | Daily step count↓ Standing%↓ Walking%↓ Sitting%↑ Sleep quality→ |
| Woodruff et al., 2021 [24] | 121 subjects (23 men, 96 women, 1 cisgender, and 1 unknown) Canada Age: 36.2 ± 13.12 years BMI: No description | Prospective observational study Between March 2020 and April 2020 | Various activity trackers, e.g., Apple Watch, Fitbit, Samsung, and Garmin | Daily step count↓ Sedentary time↑ |
| Ong et al., 2021 [26] | 1824 city-dwelling, working adults (883 men and 941 women) Singapore Age: 30.94 ± 4.62 years BMI: No description | Prospective cohort study Between 2 January 2020 and 27 April 2020 | Fitbit | Daily step count↓ Time spent on moderate-to-vigorous activity↓ Resting heart rate↓ Sleep duration↑ Sleep efficiency→ |
| Pépin et al., 2020 [27] | Approximately 742,000 individuals using wearable activity trackers (proportion of women: 37.8%) Australia, Canada, China, France, Germany, Ireland, Italy, Japan, Netherlands, Singapore, Switzerland, United Kingdom, and United States Age: 35–46 years BMI: No description | Retrospective observational study Between 1 December 2019 and 13 April 2020 | Withings | The number of steps↓ in countries with lockdown The number of steps↑ in Sweden without lockdown |
| Capodilupo and Miller, 2021 [28] | 5436 individuals using a wearable activity tracker (3900 men and 1536 women) United States Age: 40.25 ± 11.33 years BMI: No description | Retrospective observational study Between 1 January 2020 and 15 May 2020 | WHOOP strap | Exercise frequency↑ in all subjects Exercise frequency↓ in subjects aged 18–25 years Resting heart rate↓ Heart rate variability↑ Sleep duration↑ |
| Zinner et al., 2020 [29] | 14 highly trained athletes (6 men and 8 women) Germany Age: 17.1 ± 1.9 years BMI: 22.9 ± 1.4 kg/m$^2$ | Retrospective observational study During 4 weeks prior to and after the social distancing and lockdown on 23 March 2020 | Polar M430 | Training time↓ Time spent on light- and moderate-intensity physical activity↓ Sitting time↓ Time spent lying down↑ |
| Taylor et al., 2021 [30] | 311 patients with heart failure (240 men and 71 women) United Kingdom Age: 68.8 years BMI: <18.5 kg/m$^2$ (0.7%), 18.5–24.9 kg/m$^2$ (22.3%), 25–29.9 kg/m$^2$ (32.8%), >30 (44.3%) | Prospective observational study During 4 weeks preceding and following the lockdown on 23 March 2020 | Triage HF | Daily physical activity↓ |

↑, increase; ↓, decrease; BMI, body mass index; RADAR-CNS, Remote Assessment of Disease and Relapse–Central Nervous System.

It is evident that there is heterogeneity among the studies. First, the types of wearable activity trackers differ among the studies. Although there is evidence supporting inter-device validity and reliability between different types of wearable activity trackers [31], the method for measuring physical activity should be standardized. Henriksen et al. [32] developed a system to record data on physical activity from different types of wearable

activity trackers (Apple, Fitbit, Garmin, Oura, Polar, Samsung, and Withings) and confirmed that there was a significant reduction in the daily step count and energy expenditure during the lockdown period. However, the development of a more accurate and more reliable device is a challenge for the future. Second, there is a lack of information on the characteristics of subjects in the included studies. For example, five of nine studies do not show the BMI of the study subjects. In addition, most of the study subjects are probably healthy in the large-scale studies [20,26–28]; however, it is possible that those studies include subjects with chronic diseases, and the impact of the pandemic on daily physical activity differs between healthy individuals and patients with chronic diseases. Further studies assessing the change in daily physical activity in patients with chronic diseases such as diabetes, chronic kidney disease, and cancers during pandemics are warranted.

## 3. Future Perspectives of Wearable Devices during Pandemics

Wearable technologies for monitoring patients during the COVID-19 pandemic include the following: (1) respiratory rate, respiratory airflow sensing thermistor, humidity/carbon dioxide sensor, chest wall movement detection by strain/triboelectric/accelerometer, and derivation from cardiac signals using electrocardiography and photoplethysmography; (2) heart/lung sound, piezoelectric acoustic sensing; (3) oxygen saturation, optical sensing; (4) heart rate, electrical/capacitive sensing; (5) blood pressure, multimodal sensing; (6) body temperature, thermal sensing; and (7) cough, mechanical or piezoelectric sensing [5]. Wearable devices have been put to practical use for the management of patients with COVID-19; however, most of them are still in the process of development in terms of their validity and reliability in clinical practice. Real-time cardiovascular disease monitoring requires a highly powerful wireless communication technology for processing the large amount of data; however, currently, wearable medical devices cannot fully satisfy the real-time data requirements [33]. Furthermore, the diagnosis and instructions provided by doctors are manual processes; therefore, it is challenging to develop an efficient monitoring platform that can automatically detect an abnormality and provide prompt and accurate diagnosis based on the data from wearable devices. Arterial intelligence will play a crucial role in monitoring patients with chronic diseases as well as COVID-19 [34].

Wearable activity trackers may not be very useful in the management of patients with COVID-19; however, previous studies suggest that they make a significant contribution to lowering the risk of severe illness from COVID-19 through self-management of physical activity. In this review, wearable activity trackers were used only to measure physical activity during self-isolation, social distancing, or lockdown. In contrast, Quer et al. [35] developed a smartphone application that collects wearable activity tracker data and assessed whether self-reported symptoms and sensor data can differentiate patients with COVID-19 from other symptomatic subjects. The area under the curve for differentiating patients with COVID-19 and symptomatic subjects who were negative for COVID-19 was 0.80 (interquartile range, 0.73–0.86) when using the application, which suggests that the combination of wearable device data and clinical symptoms improves the diagnostic accuracy of COVID-19. This study is highly suggestive of a promising future of wearable devices. It is essential for clinicians to have a perspective on how to utilize wearable device data in the management of patients during the pandemic. This review has shown that the pandemic and/or non-pharmaceutical public health measures significantly reduce people's daily physical activity, which should be harmful for metabolic health. Indeed, the pandemic and/or non-pharmaceutical public health measures adversely impact patients with diabetes by physical inactivity, unhealthy diet, and limited access to healthcare [36]. In addition, obesity, of which physical inactivity is one of the major causes, increases the risk of other metabolic diseases such as type 2 diabetes and nonalcoholic fatty liver disease. A comparative risk assessment analysis in the US reported that 30% of COVID-19 hospitalizations were attributable to obesity and 21% were attributable to diabetes [37]. Thus, to increase or at least maintain daily physical activity during the

pandemic benefits not only individual health but also public health by reducing the burden on healthcare systems.

The problem is when and how to intervene in such sedentary individuals. The studies included in this review showed that the daily step count was decreased by approximately 1000–4000 steps per day during the pandemic. Considering that the risks of all-cause mortality (6–36%) and cardiovascular disease (5–21%) are reduced by 1000 daily step counts [38], early intervention is preferred. For example, sedentary individuals should receive a warning via wearable devices when the daily step count decreases by 1000 steps compared with the average number of steps at baseline. If the daily step count is decreased by 4000 steps or more, healthcare professionals should directly contact such individuals and check their health status. There are different types of intervention methods to promote physical activity (e.g., in-person, telephone, emails, text messaging, mobile applications, and social media), and among them, the smartphone-based intervention effectively improves total physical activity and daily steps in children and adolescents [39]. In addition, wearable activity trackers are recognized as useful and acceptable healthcare devices in individuals aged >50 years and with chronic disease [40]. Given that 66.9% of the world's population have mobile phones [41], the intervention via smartphone applications is the most promising method. Previous studies showed that the use of technologies such as smartphone applications and social media improves obesity, glycemic control, eating habits, and daily physical activity in patients with diabetes [42].

Ding et al. [5] proposed practical application scenarios of wearable devices with telehealth systems during pandemics. The scenarios are composed of hospital and outside hospital scenarios that involve various facilities and resources such as nursing homes, community care organizations, COVID-19 testing centers, mobile hospitals, hospitals and healthcare workers who monitor patients' condition in real-time, home monitoring systems, health checking centers, public areas, and COVID-19 contact tracing systems. The close network centered on a cyber hospital (database) is build up in the scenarios, and wearable devices play a vital role in data exchange between a cyber hospital and each facility or individual. Wearable activity trackers such as smartwatches are the keys to successful communication between individuals and a cyber hospital. To establish an effective healthcare system for pandemics based on the scenarios, it is significant to increase the prevalence and use rate of wearable devices among the general population. In a community-based cohort, the ownership and use rate of wearable activity trackers was only 19.6% [43]; thus, some public awareness activities and policies for the popularization of wearable technologies encouraging the application of wearable activity trackers in clinical practice may be needed. In addition, there are barriers to the purchase and use of wearable activity trackers such as cost and motivation. Gualtieri et al. [44] indicate that a free wearable activity tracker can facilitate behavior change to be more physically active in older adults with chronic diseases, while continued training and support are needed. A systematic review reported that the evidence on the effect of economic instruments including fiscal government policies to promote physical activity behavior change is limited [45]; however, government support is essential for the spread of wearable activity trackers and promoting physical activity. For instance, European governments distributed smartphone contact tracing applications for free to slow the spread of the COVID-19 pandemic [46]. It is desirable that governments are responsible for the spread of wearable devices in preparation for future pandemics. On the other hand, the validity and reliability of consumer-wearable activity trackers have been already assessed, and there was a high (Pearson or intraclass correlation coefficients $\geq 0.8$) validity of steps and a high inter-device reliability for steps, while sleep and energy expenditure were not measured adequately using existing wearable activity trackers [31]. However, technological improvement and innovation will advance, and the use of wearable activity trackers in daily clinical practice will become increasingly significant. We now await on the establishment of a healthcare system utilizing wearable activity trackers for preventing diseases due to physical inactivity and preparation for the next pandemic in the future.

## 4. Conclusions

In conclusion, daily physical activity was significantly decreased during the current COVID-19 pandemic. It depends on the types of non-pharmacological public health measure; however, the reduction in daily physical activity reaches approximately 10–50% of the amount of daily physical activity before the pandemic. After the current pandemic, the world may confront a significant increase in the number of individuals with metabolic disturbances, such as obesity and diabetes. It should be considered that healthcare professionals encourage people to increase (or at least maintain) daily physical activity via wearable technologies such as smartphone applications. If we construct an effective healthcare system involving wearable activity trackers, we will be able to prevent health problems due to physical inactivity and hospitals' burden during pandemics in the future.

**Funding:** This research received no external funding.

**Institutional Review Board Statement:** Not applicable.

**Informed Consent Statement:** Not applicable.

**Conflicts of Interest:** The author declares no conflict of interest.

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
