# Peer review of "Daily Physical Activity and Sleep Measured by Wearable Activity Trackers during the Coronavirus Disease 2019 Pandemic: A Lesson for Preventing Physical Inactivity during Future Pandemics"

_applsci, doi:10.3390/app11219956_

Round 1
Reviewer 1 Report
Mr. Hamasaki aimed to investigate the impact of the COVID-19 pandemic on physical activity measured using wearable activity trackers and discuss future perspectives on wearable activity trackers during pandemics.
Therefore he performed an extensive literature search in pubmed.
The manuscript is well structured and easy to understand.
Any interesting literature is included.
I would only add a graphic of the excluded works (including why they were excluded) as well as add a sentence in the end of the introduction section about the aim of the present manuscript (comparable to the abstract).
Table 1
I would add a column with the information in which country / countries this study was/were carried out.
Author Response
Mr. Hamasaki aimed to investigate the impact of the COVID-19 pandemic on physical activity measured using wearable activity trackers and discuss future perspectives on wearable activity trackers during pandemics.
Therefore, he performed an extensive literature search in pubmed.
The manuscript is well structured and easy to understand. Any interesting literature is included.
Thank you for taking the time and effort to carefully read and review my manuscript.
I would only add a graphic of the excluded works (including why they were excluded) as well as add a sentence in the end of the introduction section about the aim of the present manuscript (comparable to the abstract).
Thank you for your helpful comment.
According to your comment, the author added a flow diagram of this review. The author also added a sentence to the end of the introduction section about the aim of this review.
Table 1
I would add a column with the information in which country / countries this study was/were carried out.
According to your comment, the author added the information about country/countries in which the studies were carried out to Table 1.
Reviewer 2 Report
The idea of ​​this review is quite interesting and current, especially since the author try to highlight certain studies on physical activity (PA) measured during the Sars-Cov 19 pandemic, a pandemic that the whole society is struggling with even today.
''Wearable activity trackers'' - can these devices be owned by everyone? can any individual afford to buy this device? It is very difficult for me to believe this ..... I would like to ask the author how all the people would succeed or what would be the simplest way for individuals to allow themselves to buy these tools?
Keywords - I think it is more appropriate to appear other keywords, I do not think it is ok to find the same words as in the title of this paper.
Line 59- 60 - why didn't the author include the "review papers" in this study? did they not have relevant data for this review? what was the explanation for their non-inclusion? please detail this aspect more clearly
Why was the month of September not taken into account for this article, there are no articles on this topic? we are on October
Given that this review wanted to highlight the variable PA why the author also presents studies that are related to other measured variables, i.e sleep, etc? under these conditions the author can add the variable sleep in the title of the article because in many of the studies presented this term appears.
In conclusion, I have two very important things that should be explained by the author:
1. how we manage to monitor physical activities among the population (recording devices are expensive and I don't think anyone can afford to buy them).
2. Why do most variables (especially sleep) appear in most of the studies presented? aren't there studies only on the PA variable? this is the title....not PA and sleep, or PA and stress etc.
Author Response
The idea of ​​this review is quite interesting and current, especially since the author try to highlight certain studies on physical activity (PA) measured during the Sars-Cov 19 pandemic, a pandemic that the whole society is struggling with even today.
I thank you for your comments, which have helped me to improve the manuscript.
''Wearable activity trackers'' - can these devices be owned by everyone? can any individual afford to buy this device? It is very difficult for me to believe this ..... I would like to ask the author how all the people would succeed or what would be the simplest way for individuals to allow themselves to buy these tools?
Thank you for your important comment.
There are barriers to the purchase and use of wearable activity trackers such as cost and motivation. Gualtieri et al. [44] indicate that a free wearable activity tracker can facilitate behavior change to be more physically active in older adults with chronic diseases, while continued training and support are needed. As the author mentioned in the manuscript, considering that 66.9% of the world’s population have mobile phones, the intervention via smartphone applications is feasible and cost-effective. A systematic review reported that the evidence on the effect of economic instruments including fiscal government policies to promote physical activity behavior change is limited [45]; however, government support is essential for the spread of wearable activity trackers and promoting physical activity. For instance, European governments distributed smartphone contact tracing applications for free to slow the spread of the COVID-19 pandemic [46]. It is desirable that governments are responsible for the spread of wearable devices in preparation for future pandemics.
The author added these sentences to the section of Future perspectives of wearable devices during pandemics (line 327-339).
Keywords - I think it is more appropriate to appear other keywords, I do not think it is ok to find the same words as in the title of this paper.
According to your comment, the author changed the keywords as follows: wearable device; smartphone application; telemedicine; physical activity; COVID-19
Line 59- 60 - why didn't the author include the "review papers" in this study? did they not have relevant data for this review? what was the explanation for their non-inclusion? please detail this aspect more clearly
There are no review papers that summarize the relationship between physical activity measured by using wearable activity trackers and the COVID-19 pandemic. In accordance with your comment, the author deleted the term “review papers” from the exclusion criteria to avoid a misunderstanding.
Why was the month of September not taken into account for this article, there are no articles on this topic? we are on October.
The author performed the literature search on the end of August. It took a month to review the relevant studies and write up the manuscript; thus, the author could not include the studies that were published in and after September. Unfortunately, it is a matter of time. According to your comment, the author performed the literature search of PubMed again from its inception to date (20 October 2021), and it yielded 294 articles. Of these articles, one study that was published on 12 October investigated how COVID-19 lockdown affected sleep, physical activity, and wellbeing by using a wearable sleep/activity tracker [Sleep. 2021 Oct 12:zsab250. doi: 10.1093/sleep/zsab250.]. This study indicates that the re-opening after lockdown is associated with an increase of physical activity. However, the finding of this study does not influence the conclusion of this review.
Given that this review wanted to highlight the variable PA why the author also presents studies that are related to other measured variables, i.e sleep, etc? under these conditions the author can add the variable sleep in the title of the article because in many of the studies presented this term appears.
As you pointed out, most of the included studies assessed the impact of the pandemic on sleep as well as physical activity. According to your comment, the author changed the title to “Daily physical activity and sleep measured by wearable activity trackers during the coronavirus disease 2019 pandemic: a lesson for preventing physical inactivity during future pandemics.”
In conclusion, I have two very important things that should be explained by the author:
- how we manage to monitor physical activities among the population (recording devices are expensive and I don't think anyone can afford to buy them).
Wearable activity trackers especially smartphone applications should be distributed to as many people as possible under government support for individuals.
- Why do most variables (especially sleep) appear in most of the studies presented? aren't there studies only on the PA variable? this is the title....not PA and sleep, or PA and stress etc.
In most of the included studies, sleep (e.g., duration and quality) as well as physical activity was included as a primary outcome. The title of this review was changed accordingly.
I would appreciate it if you would check the revised version of my manuscript.
Round 2
Reviewer 2 Report
-